

# 1    Organic Pollutant Oxidation on Manganese Oxides in Soils - The

# 2    Role of Calcite Indicated by Geoelectrical and Chemical Analyses

Sonya S. Altzitser [1], Yael G. Mishael [1], Nimrod Schwartz [1]
[1] Department of Soil and Water Sciences, The Robert H. Smith Faculty of Agriculture, Food and Environment, The
Hebrew University of Jerusalem, Rehovot, 7610001, Israel
*Correspondence to:* Nimrod Schwartz (Nimrod.schwartz@mail.huji.ac.il)



**Abstract.** Understanding phenolic pollutants interaction with soil colloids has been a focus of extensive research, primarily under controlled conditions. This study addresses the need to explore these processes in a more natural, complex soil environment. We aim to enlighten the underlying mechanisms of hydroquinone (a representative phenolic pollutant) oxidation in ambient, $MnO_2$-rich sandy soil within soil columns designed for breakthrough experiments. Our innovative approach combines noninvasive electrical measurements, crystallographic and microscopic analyses, and chemical profiling to comprehensively understand soil-pollutant interactions. Our study reveals that hydroquinone oxidation by $MnO_2$ initiates a cascade of reactions, altering local pH, calcite dissolution, and precipitating amorphous Mn-oxides, showcasing a complex interplay of chemical processes. Our analysis, combining insights from chemistry and electrical measurements, reveals the oxidation process led to a constant decrease in polarizing surfaces, as indicated by quadrature conductivity monitoring. Furthermore, dynamic shifts in the soil solution chemistry (changes in the calcium and manganese concentrations, pH, and EC) correlated with the non-monotonous behavior of the in-phase conductivity. Our findings conclusively demonstrate that the noninvasive electrical method allows real-time monitoring of calcite dissolution, serving as a direct cursor to the oxidation process of hydroquinone, enabling the observation of soil surface processes, and chemical interactions.



## 1. Introduction

Phenolic pollutants may originate from various sources, including agricultural, industrial, municipal, and medical wastes (Davì and Gnudi, 1999; Farhan Hanafi and Sapawe, 2020). Due to their chemical characteristics, phenolic pollutants tend to persist in soil and water at relatively low concentrations for an extended period, posing a significant environmental threat. The chemical fate of these pollutants in soil has been extensively studied, with particular attention given to processes such as adsorption-desorption and oxidation (Ahmed et al., 2015; Delgado-Moreno et al., 2021; Kang and Choi, 2008; Lambert, 2018; Loffredo and Senesi, 2006; Sun et al., 2022). Regarding oxidation, various oxides, both natural and engineered, have been investigated for their potential to remove phenolic pollutants (Gusain et al., 2019; Remucal and Ginder-Vogel, 2014). Among these, birnessite ($MnO_2$), a manganese oxide naturally found in soils, is known for its effectiveness in oxidizing various phenolic compounds (Murray, 1974; Remucal and Ginder-Vogel, 2014).

While manganese oxides' ability to oxidize phenols has been explored widely, most of these studies have been conducted in buffered, controlled environments within batch experiments, which may not accurately reflect manganese oxide behavior in more complex, heterogeneous soil (Chien et al., 2009; Fukuzumi et al., 1975; Liao et al., 2021; Liu et al., 2011; McBride, 1987; McKenzie, 1971; Shindo and Huang, 1984; Stone and H, 1989; Trainer et al., 2021). To the best of our knowledge, only a few works have investigated the oxidation of phenolic pollutants by manganese oxides in situ in soils. For instance, studies have shown the oxidation of phenolic acids and dissolved organic matter by manganese oxides in soils, but these works focused on naturally occurring, non-contaminating compounds rather than phenolic pollutant fate in the soil (Ding et al., 2022; Lehmann et al., 1987). In a study by Grebel et al. (Grebel et al., 2016), the oxidation of various phenolic contaminants was investigated using engineered $MnO_2$-coated sand columns, and their key conclusion underscores $MnO_2$ efficacy as an oxidizing agent for phenolic contaminants. However, to determine whether $MnO_2$ can be equally effective in natural soil environments, further investigation is required.

We aim to investigate the fate of phenolic pollutants, specifically in the context of oxidation processes in $MnO_2$-enriched soil To achieve this, we will apply both classical methodologies and an advanced geoelectrical method recently introduced to soil science: spectral induced polarization (SIP) (Gao et al., 2019; Johansson et al., 2019; Mellage et al., 2022; Revil, 2012; Schwartz et al., 2012; Schwartz and Furman, 2012; Shefer et al., 2013; Vaudelet et al., 2011; Vinegar and Waxman, 2012; Zhang et al., 2012). This approach allows us not only to track the transformation of phenolic pollutants through oxidation by $MnO_2$ but also to monitor the broader impacts of this oxidation process on other elements within the soil environment.

SIP is a method where a low frequency, time dependent electrical field is applied, and the resultant potential is recorded. This technique captures both the conductive and capacitive characteristics of the subsurface, characterized by the in-phase ($\sigma'$) and quadrature ($\sigma''$) conductivity, respectively, in a non-invasive way (Binley and Kemna, 2005; Reynolds, 2011). Quadrature and in-phase conductivity are associated with the interfacial chemistry of the grain surface and grain size, while in-phase conductivity is also related to pore-water electrolyte conductivity (Ben Moshe and Furman, 2022). The $\sigma''$ is frequency dependent and related to polarization processes at the electric double layer



(EDL), and indeed Vinegar & Waxman (Vinegar and Waxman, 2012) proposed a linear relationship between the soil
cation exchange capacity (CEC) and the $\sigma''$. Additionally, studies on the impact of organic contaminants on the low-
frequency complex conductivity of soils and porous materials demonstrated the ability of the SIP method to detect
and monitor organic contaminants within the subsurface (Kirmizakis et al., 2020; Mellage et al., 2018, 2022; Revil,
2012; Schwartz et al., 2020; Schwartz and Furman, 2012, 2015; Vaudelet et al., 2011).
This study aims to thoroughly explore the behavior of hydroquinone, a model phenolic molecule with a well-known
oxidation mechanism by Mn-oxides, in $MnO_2$-enriched sandy soil (Mn-sandy soil). To achieve this, we employed an
array of methods including; electric measurements of the soil profile, crystallographic and microscopic examination
of the soil minerals, and chemical analysis of the soil solution. We hypothesized that integrating electrical
measurements, soil solution analysis, and soil surface examinations would enable us to reach a unique understanding
of the oxidation process in the soil and provide insights into the resulting chemical mechanisms in the soil
environment.
**2.   Materials and Methods**
In this study, we investigated the oxidation of hydroquinone by $MnO_2$ in sandy soil column experiments. The
experiments were conducted using sandy soil and $MnO_2$-enriched sandy soil (Mn-sandy soil) to observe the behavior
of hydroquinone and its oxidation product, benzoquinone. During the experiments, we employed SIP measurements
to study the electrical characteristics of the soil as the oxidation process occurred. We analyzed the samples for
hydroquinone and benzoquinone concentrations using High-Performance Liquid Chromatography (HPLC), and
measured ion concentrations and composition by Coupled Plasma Atomic Emission Spectrometer (ICP-AES), pH,
and Electrical Conductivity (EC). Additionally, we conducted Scanning Electron Microscopy (SEM), Energy-
Dispersive X-ray Spectroscopy (EDS), and X-ray Diffraction (XRD) analyses to observe any changes in soil
morphology and mineralogy before and after the introduction of hydroquinone to the soil.
**2.1. Materials**
Hydroquinone (99% purity), benzoquinone (99% purity), acetonitrile (HPLC grade), and calcium chloride were
purchased from Sigma-Aldrich. Potassium permanganate, and hydrochloride acid 32% were purchased from Mercury
LTD. Sandy soil with 97% sand and 3% silt (measured using PRIO, Meter group, Germany), contains 4% $CaCO_3$, and
2.5% organic matter.
**2.2. $MnO_2$ preparation**
$MnO_2$ was synthesized following the procedure of McKenzie (McKenzie, 1971). In brief, concentrated HCl was added
dropwise to a boiling solution of potassium permanganate to form a dark purple precipitate of $\delta$-$MnO_2$. After synthesis,
the suspension was centrifuged (15,200 g, 15 min), and the supernatant was decanted and replaced with double-
deionized water. The procedure was repeated until the supernatant was colorless; then the slurry was oven-dried
overnight at 35 °C and freeze-dried.



### 2.3. Methods

#### 2.3.1. Spectral induced polarization measurements

In the SIP method, a low frequency (typically 0.01 Hz to 10 kHz) oscillating current I (A) is applied through two electrodes on a porous medium, and electrical potential U (V) is measured by two other electrodes. Using Ohm's law, the complex admittance of the medium, $Y^* = I/U$ (S) is obtained. The complex conductivity is related to the admittance through the geometric factor G ($m^{-1}$) such that $\sigma^* = G \cdot Y^*$. The complex conductivity signal can be expressed as $\sigma^* = \sigma' + i\sigma'' = |\sigma^*|e^{i\varphi}$, where $\sigma'$ (S m$^{-1}$) is the in-phase conductivity, associated with energy dissipation processes, $\sigma''$ (S m$^{-1}$) is the quadrature conductivity, related to energy storage processes (Schwartz and Furman, 2012), and $\varphi$ (rad) is the phase shift.

The SIP signal was measured using the PSIP impedance spectrometer (Ontash & Ermac Inc, NJ, USA), in polyvinyl chloride (3 cm diameter, 30 cm long) columns equipped with 4 brass electrodes, 6 mm in diameter, for both current injection and potential measurement (Fig.1). The current electrodes were 8 cm long and they crossed the entire sample, while the potential electrodes were 5 cm long, and they were retraced in their respective holes to prevent electrode polarization (as suggested by Cassiani et.al (Cassiani et al., 2009), and Schwartz and Furman (Schwartz and Furman, 2012). Electrical contact between the potential electrodes and the sample was ensured through the electrolyte. The geometric factor (G) was determined by measuring the admittance of a series of electrolytes with different electrical conductivities.

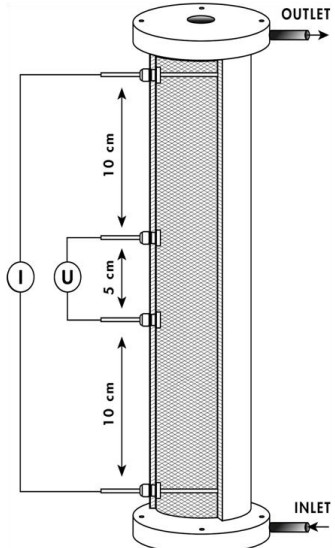

*Figure 1. Scheme of experimental SIP column: inlet solution is injected in through the bottom and the outlet is collected in fractions. The current (I) is injected between the top and the bottom electrodes, and the SIP signal is measured between the potential electrodes (U)*



### 2.3.2. Column experiments

Breakthrough experiments were conducted to study the behavior of hydroquinone, and benzoquinone in sandy soil and Mn-sandy soil in an unbuffered environment. Two sets of experiments were performed using triplicate columns for each treatment: untreated sandy soil and Mn-sandy soil, which was prepared by mixing sandy soil with 5% w/w $MnO_2$. Air-dried soil was mixed with 10% w/w of $CaCl_2$ 5mM solution as the saturating liquid. This soil was then packed in the columns in approximately 100 mL increments and gently compressed after each addition. Based on an assumed particle density of 2.65 g cm$^{-3}$ (Warrick, 2002), the average porosity of the sandy soil and Mn-sandy soil samples was 0.4 and 0.44±0.02, respectively.

After packing, the columns were placed vertically and a 5mM $CaCl_2$ solution was introduced from the bottom to wash away excess salt, ensuring saturated flow at a constant flow rate of 1 mL/min using a peristaltic pump (Masterflex L/S series, Cole-Parmer Inc., IL, USA). The soil was washed with $CaCl_2$ until equilibrium was reached between the inlet and outlet solutions (EC=900 µS cm$^{-1}$). Upon reaching equilibrium, the inlet solution was replaced by a mixed solution of hydroquinone and benzoquinone (0.1M each in $CaCl_2$ 5mM solution), or by a hydroquinone solution (0.1M in $CaCl_2$ 5mM solution) passing through the sandy soil and Mn-sandy soil columns for 4 or 8 pore volumes (PV), respectively, until mass balance was achieved. Both solutions were left unbuffered and unpurged to better represent natural conditions.

Throughout the experiments, continuous SIP measurements were taken, and at 20-minute intervals, 2 mL samples of the outlet solution were collected and immediately filtered using a 0.22 µm reverse cellulose membrane filter syringe for further analysis. The collected outflow was analyzed to determine (1) hydroquinone and benzoquinone concentrations using HPLC (Waters 600, Waters, Milford, MA), equipped with a diode-array detector. The HPLC column was an XBridge Phenyl 3.5 µm 4.6X150 mm, with a flow rate of 1 mL/min, and the column temperature was set to 25 °C. Hydroquinone and benzoquinone were monitored at wavelengths of 222 nm and 246 nm, respectively. The mobile phase consisted of acetonitrile and double distilled water (DDW). The phase gradient started at 5% acetonitrile for 0-3 min, linearly increased to 40% for 3-10 min, and then increased again to 95% over 10-11 min. Acetonitrile maintained at 95% over 11-12 min, then decreased back to 5% over 12-13 min, and maintained at 5% for 13-16 min. (2) $Ca^{2+}$ and soluble Mn concentration in the effluent by ICP-AES (Arcos Spectro Ltd., Germany), and (3) pH and EC values.

### 2.3.3. Colloid surface analysis by SEM-EDS and XRD

Sandy soil, $MnO_2$, and Mn-sandy soil morphology was observed, before and after the introduction of hydroquinone by SEM (JEOL IT 100 Low vacuum). All samples were oven-dried at 40°C and thinly ground before analysis, mounted on 30 mm round SEM aluminum stubs using adhesive carbon tape. Secondary electron images were taken using the following operating conditions: 20 keV, 9 mm WD, and x350 magnification for all samples. For each soil, 10 images were obtained and scanned for calcium, manganese, and silica semi-quantitative percentages, using EDS. The concentration of the elements in the Mn-sandy soil samples was corrected to the relative addition of Mn to the system. The effect of hydroquinone oxidation on Ca and Mn content in the soil was conducted using a non-parametric



comparison for each pair, using the Wilcoxon method. The statistical analysis was carried out by JMP®, Version 16.
SAS Institute Inc.
The mineralogy of the soil and the change in $MnO_2$ mineralogy, pre- and post-oxidation were also evaluated by XRD.
Soil samples were ground and loaded into an XRD sample holder by front loading followed by razor blade leveling.
XRD patterns were acquired in Bragg-Brentano geometry using a PANalytical X'Pert diffractometer with CuKα
radiation operated at 45 kV and 40 mA. The samples were scanned from 5 to 70° 2θ at a step size of 0.013° 2θ, using
a PIXcel detector in continuous scanning line (1D) mode with an active length of 3.35°. Mineral phase identification
was performed using HighScore Plus® software based on the ICSD database.
### 3. Results and Discussion
#### 3.1. Hydroquinone and benzoquinone fate in sandy soils - breakthrough curves
Figure 2 illustrates the breakthrough curves of hydroquinone and benzoquinone in sandy soil (Fig. 2a) and Mn-sandy
soil (Fig. 2b) columns. In the control sandy soil columns (Fig. 2a), both hydroquinone and benzoquinone exhibited
classic symmetric sigmoidal breakthrough curves, with the breakthrough occurring at approximately 1 pore volume
(PV). This suggests that there was negligible adsorption or chemical transformation of these compounds in the sandy
soil, allowing them to pass through the column relatively unimpeded. In contrast, the breakthrough curves in the Mn-
sandy soil columns (Fig. 2b) demonstrate different behavior. Benzoquinone showed an initial breakthrough at around
4 PVs, reaching a relative concentration ($C/C_0$) of about 0.2, and continued to increase gradually. Hydroquinone,
however, exhibited a significantly delayed breakthrough, occurring at approximately 7 PVs with a relative
concentration of 0.7. The moderate slopes of these breakthrough curves compared to the steep slopes observed in the
sandy soil columns indicate that hydroquinone undergoes oxidation in the presence of $MnO2$, forming benzoquinone.
This oxidation process is responsible for the delayed and more gradual breakthrough of hydroquinone, highlighting
the reactive nature of the Mn-sandy soil in altering the transport and fate of these pollutants (Buamah et al., 2009).

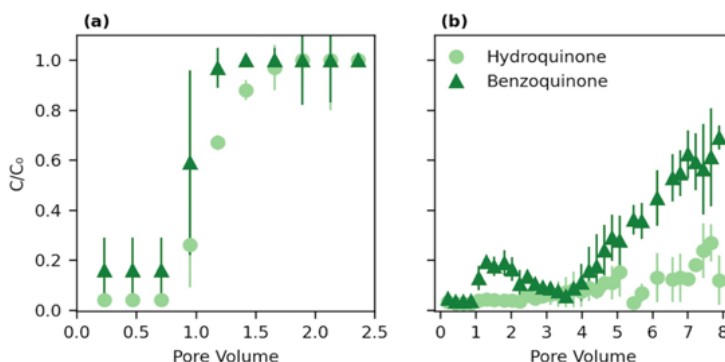

*Figure 2. Hydroquinone (0.1 M initial) and benzoquinone (0.1M) relative concentrations, in (a) sandy soil and (b) Mn-sandy soil columns*




### 3.2. SIP and soil solution chemistry monitoring


Sandy soil (control) and Mn-sandy soil columns were saturated with a background solution (5 mM CaCl$_2$) and their
SIP signatures were recorded upon reaching equilibrium, before the introduction of the pollutant (Fig. 3).

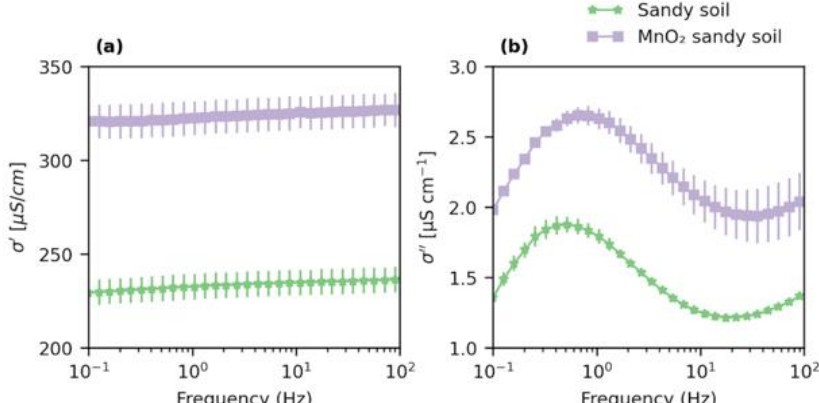

*Figure 3. In-phase ($\sigma'$) (a) and quadrature ($\sigma''$) (b) conductivity of sandy and Mn-sandy soils (5 % w/w).*


The quadrature conductivity ($\sigma''$) (associated with surface polarization) of the sandy soil, exhibited classical spectra
for frequency-dependent polarization of porous media, with a peak at around 0.5 Hz (Fig. 3**Error! Reference source**
**not found.**b). Compared to the sandy soil, $\sigma''$ of Mn- sandy soil increased by $\sim$ 40%, most likely due to the
contribution of the high CEC of MnO$_2$ (Händel et al., 2013; McKenzie, 1971; Post, 1999). Similarly, the in-phase
conductivity $\sigma'$ was also higher ($\sim$ 40%) than that of the sandy soil (Fig. 3a), most probably due to the contribution
of MnO$_2$ to the surface conductivity of the media (recall, that the EC of the soil solution was kept constant between
the treatments). Upon the addition of hydroquinone, the control columns demonstrated no change in both $\sigma''$ and $\sigma'$
throughout the experiment (Fig. 4a, b). This was accompanied by negligible benzoquinone, Ca$^{2+}$, and manganese
concentrations measured in the effluent. Additionally, the EC remained constant at 1 mS m$^{-1}$ $\pm$0.3 and the pH values
were steady at 9$\pm$0.2 (Fig 5. a, b). These results indicate that there was neither adsorption nor oxidation of
hydroquinone in the control columns, as also demonstrated by the breakthrough curves (Figure 2a).
Hydroquinone flows through Mn-sandy soil columns induced a constant decrease in $\sigma''$, as expected, due to oxidation
processes in the system, resulting in a reduction in oxidizing and polarizing surfaces (Fig. 4 c, e). On the other hand,
the $\sigma'$ increased up to ~4 PVs and then decreased (Fig. 4 d, f). The maximum $\sigma'$ value reached post ~4 PVs
corresponded with peaks in EC and pH values, as well as Ca$^{2+}$ concentration in the effluents (Fig 5. c, d). The pH
value and EC dramatically increased from 8.82 ($\pm$0.25) and 1.7 mS m$^{-1}$ ($\pm$0.4) to 10.8 ($\pm$0.1) and 4.53 mS m$^{-1}$ ($\pm$0.03),
respectively (Fig. 5 c). Simultaneously, the Ca$^{2+}$ concentrations increased noticeably (from 0.1 to 25 mM) while the
manganese concentrations increased only slightly (from below the detection limit to 0.1 mM) (Fig 5. d). Indeed, $\sigma'$ is
related to the bulk solution properties, i.e., an increase in ion concentration, mainly Ca$^{2+}$, will result in an increase in



$\sigma'$. Notably, all maxima for $Ca^{2+}$, EC, $\sigma'$, and pH corresponded with hydroquinone oxidation, as shown by benzoquinone breakthrough (Fig. 2b). Since these trends are not observed in the control sandy soil columns (with hydroquinone flow, but without $MnO_2$) we suggest that hydroquinone oxidation by $MnO_2$ surfaces initiated a cascade of reactions: (i) a local increase in proton concentration due to hydroquinone deprotonation decreasing the local pH (Rudolph et al., 2013). At this stage two reactions, which require protons, may take place simultaneously but at different rates: (iia) calcite dissolution (evident by the $Ca^{2+}$, EC, $\sigma'$, and pH measurements), and (iib) $MnO_2$ reduction and dissolution to $Mn^{2+/3+}$ (Fukuzumi et al., 1975; McBride, 1987; Remucal and Ginder-Vogel, 2014; Stone and H, 1989), evident by benzoquinone breakthrough. The kinetics of calcite acid dissolution is at least 8 orders of magnitude higher than oxide acid dissolution (Anon, 2004) i.e., the protons are consumed faster by the calcite than by the oxidation reaction. (iii) The oxidation processes diminish, due to adsorption or precipitation of $Mn^{2+/3+}$ as amorphous Mn-oxides on the birnessite ($MnO_2$) surface (Ding et al., 2022; Remucal and Ginder-Vogel, 2014; Stone and H, 1989), also supported by the very low manganese concentrations eluting (0.1 mM). (iv) Calcite dissolution is suppressed, resulting in a decrease in $Ca^{2+}$, EC, $\sigma'$, and pH. Indeed, the decrease in $\sigma''$ reflects the reduction in $CaCO_3$ content in the soil (Izumoto et al., 2020; Wu et al., 2010) and may also correlate to a reduction in active $MnO_2$ surfaces. To further support this suggested cascade of reactions, we tested the precipitation of $Mn^{+2/+3}$ as amorphous Mn-oxides.





205

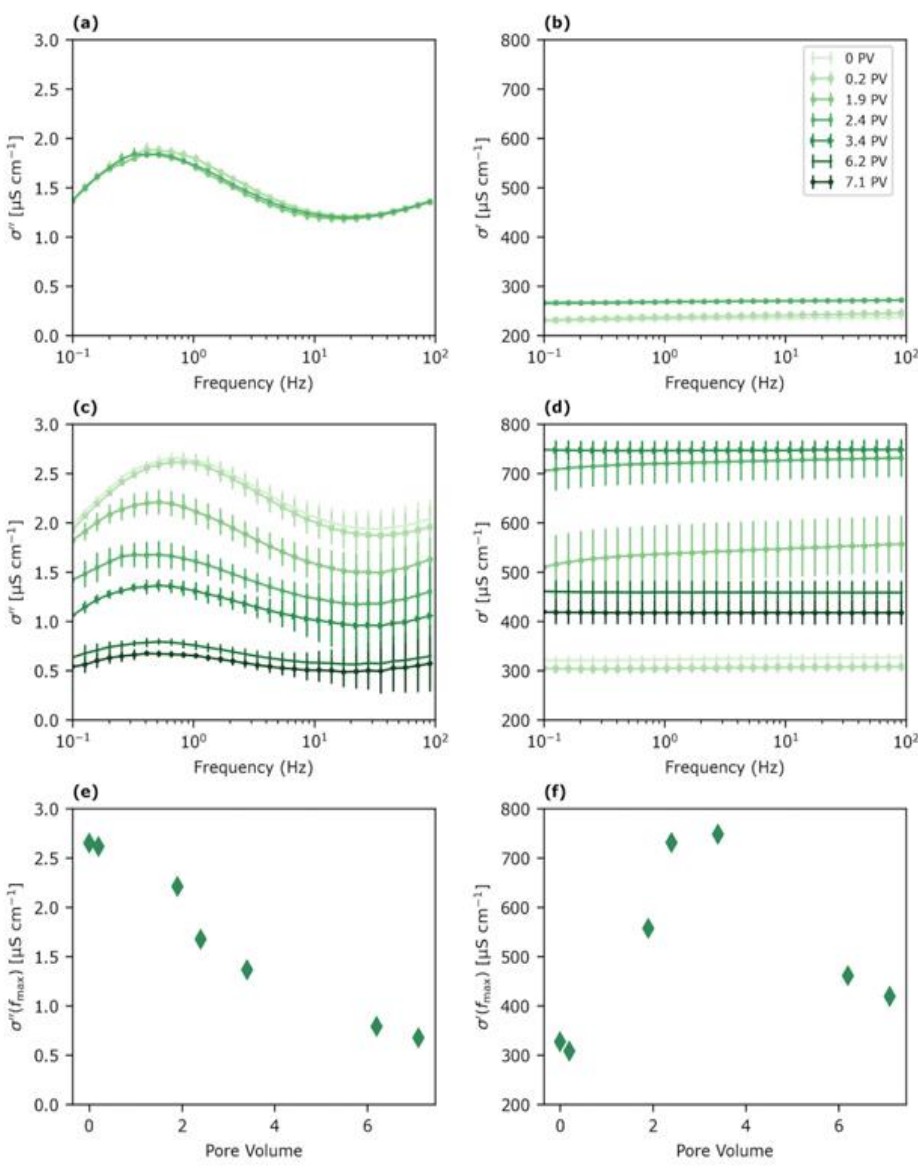

*Figure 4. Quadrature ($\sigma''$) and in-phase ($\sigma'$) conductivity of sandy (a,b) and Mn-sandy soils (c,d) during hydroquinone oxidation. $\sigma''$ (e) and $\sigma'$ (f) at the peak frequency of Mn-sandy soil*

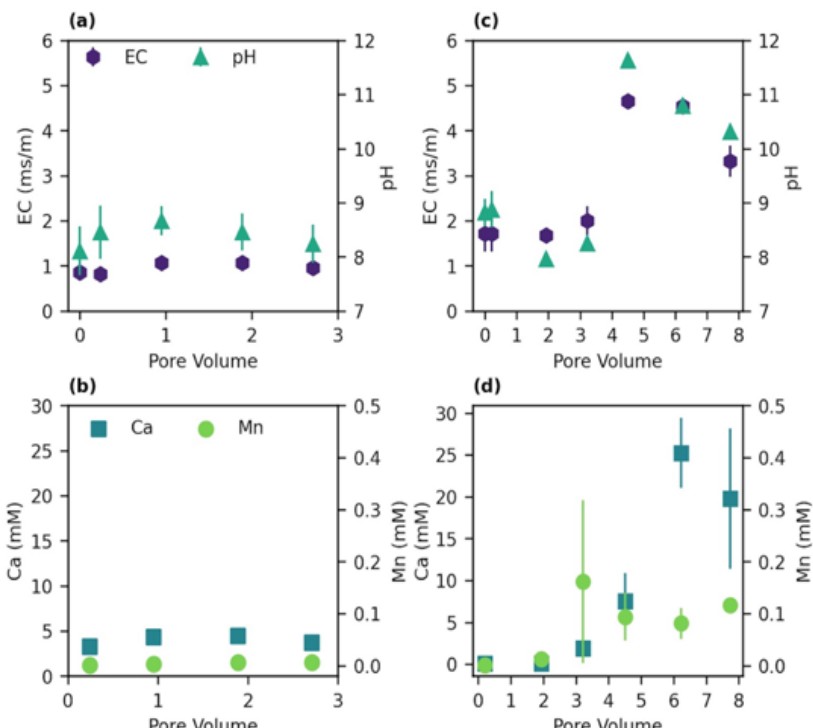

*Figure 5. Effluent measurements of EC, pH, $Ca^{2+}$ and Mn concentrations in (a), (b) sandy and (c), (d) Mn-sandy soil*

### 3.3. Soil mineral characterization

To further test $MnO_2$ dissolution and $Mn^{2+/3+}$ precipitation, we characterized the soil samples before and after hydroquinone oxidation using XRD (Fig. 6a). The sandy soil was found to be composed mainly of quartz, feldspar, and calcite. X-ray diffractogram analysis confirmed that the Mn-sandy soil initially contained approximately 5% of $MnO_2$. After hydroquinone oxidation, the $MnO_2$ content was reduced to ~ 1%, indicating that $Mn^{4+}$ was most likely reduced to $Mn^{2+/3+}$. These reduced manganese ions likely precipitated as amorphous Mn-oxides or were adsorbed onto the $MnO_2$ surface, which would not be detected by XRD. This conclusion is further supported by the very low concentration of $Mn^{2+/3+}$ eluting from the columns (Fig. 5b, d).

Finally, SEM images coupled with EDS analysis of the samples (2 replicates, 10 images per sample) confirm the reduction in Ca content post oxidation, while the Mn content remains constant in both samples (Fig. 6 b). SEM images vividly depict the morphology of pure MnO2 (Fig. 6c), quartz, and CaCO3 deposits in the sandy soil samples (Fig. 6d). In the Mn-sandy soil samples, $MnO_2$ is also notably present (Fig. 6e). Comparisons of post-oxidation samples to pre-oxidation samples showed no significant visual changes (Fig 6. e, f).



221

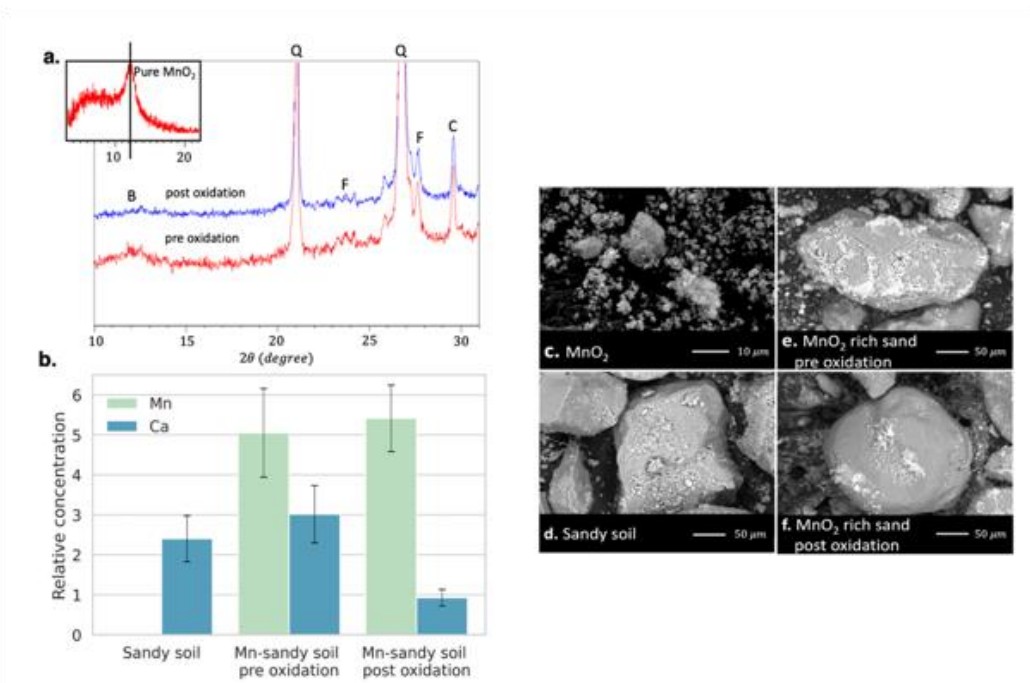

*Figure 6. (a) X-ray diffraction full characterization curve of Mn-sandy soil, pre and post oxidation, and MnO₂ inset. B, Q, F, and C represents the peaks of birnessite, quartz, feldspar, and calcite (b) Ca and Mn relative concentration by EDS analysis, and scanning electron micrographs of (c) MnO₂, (d) sandy soil, (e) MnO₂ rich sandy soil, and (f) MnO₂ rich sandy soil post oxidation.*

## 4. Conclusion

We explored the oxidation mechanism of hydroquinone in Mn-sandy soil by applying a combination of methods, including electrical measurements (SIP), crystallographic analysis (XRD), microscopic examination (SEM-EDS) of soil minerals, and chemical assessments of the soil solution (HPLC, ICP, pH, EC, etc.). Integrating results from these different methods provided insights into subsequent reactions such as mineral dissolution, chemical precipitation, and leaching.

Our findings suggest that hydroquinone oxidation by $MnO_2$ surfaces to benzoquinone initiated a cascade of reactions starting with local alterations in pH levels. These triggers increased $CaCO_3$ dissolution, while simultaneously, $MnO_2$ reduction results in its dissolution to $Mn^{2+/3+}$ and adsorption or precipitation as amorphous Mn-oxides on the $MnO_2$ surface. These results were supported by both chemical and electrical measurements. $CaCO_3$ dissolution was identified by a reduction in Ca by SEM-EDS analysis and by the SIP signature, showing a consistent decline in $\sigma''$ due to a reduction in polarized surfaces. The changes in $Ca^{2+}$ concentrations in the effluent were monitored by ICP and reflected by the alternating σ' signature. The effluent contained only minimal Mn concentration. XRD and SEM-EDS analysis results demonstrated a reduction in $MnO_2$ content and constant Mn content, respectively.



These combined findings support the precipitation of $Mn^{2+/3+}$ as amorphous Mn-oxides and $MnO_2$ surface passivation,
most likely also contributing to the consistent decrease in $\sigma''$. This study has provided valuable insights into the
sensitivity of SIP signatures to changes in soil properties, due to oxidation processes within the soil. Future research
should include the dynamic role of microbial activity in altering soil redox conditions, leading to $MnO_2$ reduction or
$CaCO_3$ dissolution. Furthermore, deeper exploration into the implications of soil structure changes resulting from
$CaCO_3$ dissolution and precipitation on the fate of pollutants in the subsurface is necessary, considering diverse
pollutant groups, organic matter, etc. In natural environments and at field scale, the complexities will require further
investigation, potentially formulating effective environmental remediation strategies.

**Competing interests**
The contact author has declared that none of the authors has any competing interests
**Acknowledgments**
This research was supported by grant 80689 from the Ministry of Innovation, Science and Technology of Israel. We
also acknowledge The Hebrew University of Jerusalem for internal funding.

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
