# Peer review of "Organic Pollutant Oxidation on Manganese Oxides in Soils The"

_EGUsphere, 2024_

## Author Comment (AC1)

*Response to reviewer comment 1 | Egusphere 2024-2101*

This manuscript submitted by Altzitser et al. for publication in the journal SOIL present a very clear and convincing study that uses geophysics to look at and better understand oxidation processes of an organic pollutant. The authors use a state-of-the-art experimental set-up to measure the Spectral Induced Polarization (SIP) signature of this process in well-controlled laboratory conditions. Their experimental results are clear and unambiguous, showing that SIP shows strong potential to non-intrusively monitor this oxidation process. I have a couple of minor comments on the manuscript that I develop in the list below. However, after these small changes, I will be happy to recommend the publication of this manuscript in SOIL.

*We sincerely thank the reviewer for the constructive feedback. We have carefully considered all the comments and will revise the manuscript accordingly. Below, we provide detailed responses to each point raised:*

General comment: The resolution of the figures on the pdf is rather poor, I guess it is a conversion issue but it would be good to enhance their resolution (especially for the pictures on Fig. 6).

*The resolution issue arose due to file size constraints during the PDF conversion process. We will address this by ensuring that all images, particularly those in Fig. 6, are submitted in high resolution for the final version to maintain clarity.*

Detailed comments:

Line 46-48: Given the context, I suggest to cite Kessouri et al. (2019). Also, note that Revil et al. (2021) is dedicated to the use of SIP on soils.

*We have added the suggested citations of Kessouri et al. (2019) and Revil et al. (2021).*

Line 53: Note that Binley and Slater (2020) is more recent book reference.

*Thank you for noticing. We have added Binley and Slater (2020) as a citation in the relevant section to reflect the most recent reference.*

Subsection 2.1: Since, sand technically refer to a grain size, it would be more complete to provide the mineral constituting sand and silt grains.

*In our study, we use 'Sandy soil' to describe the soil texture, which is a common practice in soil science to indicate the predominant particle size. We provided the percentage of sand (97%), silt (3%), organic matter (OM)(2.5%), and CaCO₃(4%) to give additional context about the soil composition.*

Section 3: In the text, it could help the reader to illustrate more explicitly the chemical reactions.

*Thank you for the suggestion, we added the following chemical reaction equation to Figure 2.*

Figure 3: the author should homogenize their notation, here the units on the two y-axes could be written following the same convention (later the authors use "cm⁻¹" rather than "/cm", I suggest to keep it everywhere).

*Thank you for noticing. We ensured that the notation is homogenized, using "$cm^{-1}$" consistently throughout the manuscript.*

Line 172: Reference problem.

*The reference issue at Line 172 has been fixed.*

Line 189: Note that a concentration increase of one ion does not always induce an increase of electrical conductivity. Indeed, as shown by Rembert et al. (2021) replacing the very mobile H+ ions with the "heavier" hydrated Ca2+ ions tends to decrease the water electrical conductivity during calcite dissolution (their Fig. 4 and discussion). Hence, it is rather the complete reaction that can explain this change than only its product.

*We appreciate the reviewer's insightful comment and the reference to Rembert et al. (2021). We agree that the relationship between ion concentrations and electrical conductivity is complex and that replacing highly mobile ions (like $H^+$) with less mobile ones (like $Ca^{2+}$) can sometimes decrease conductivity, as shown in their study.*

*In our specific case, however, we believe the concentration effects dominate due to the following reasons:*

1. *The $H^+$ concentration in our system ranges from $10^{-8}$ to $10^{-12}$ M, while $Ca^{2+}$ and $Mn^{2+}$ concentrations are in the $10^{-3}$ M range. This significant difference in concentration (5-9 orders of magnitude) means that changes in the major ion concentrations have a more substantial effect on conductivity than $H^+$ replacement.*

2. *The mobilities of $Ca^{2+}$ and $Mn^{2+}$ are relatively similar ($Ca^{2+}$: 6.17 x $10^{-8}$ $m^2$ $Vs^{-1}$, $Mn^{2+}$: 5.5 x $10^{-8}$ $m^2$ $Vs^{-1}$ at 25°C), so replacing one with the other doesn't significantly affect overall mobility. While $H^+$ ions are indeed much more mobile (36.23 x $10^{-8}$ $m^2$ $Vs^{-1}$ at 25°C), their extremely low concentration in our system means their contribution to overall conductivity is limited.*

*Given these conditions, an increase in the concentration of one of the major ions ($Ca^{2+}$ or $Mn^{2+}$) is expected to increase the in-phase conductivity in our system. However, we acknowledge that in systems with different ionic compositions or concentration ranges, the interplay between ion replacement and conductivity could be more complex, as demonstrated by Rembert et al. (2021).*

Figure 5: On 5a and b, the unit should be written with a capital "S". Also why not using the same unit as for the previous figures (i.e., μS cm-1)?

*Thank you for noticing. We corrected the unit notation to use a capital "S" and ensure consistency by using the unit μS $cm^{-1}$, as in the previous figures.*

---

## Author Comment (AC2)

The manuscript by Altzitset et al. presents results from an experimental investigation where spectral induced polarization was applied to monitor hydroquinone oxidation in the presence of MnO2 in an artificial soil mixture of a sandy soil amended with MnO2. The findings are convincing and the quality of the experiments provides confidence in the results. However, the authors provide only minimal insight into the chemical reactions occurring in the system that support their conceptual model. In addition, the analysis of breakthrough curves remains qualitative, and would (potentially) benefit from mass balance calculations to better justify the findings in the solid phase characterization. Based on these general remarks and my specific comments below I recommend moderate revisions before this paper can be accepted.

*We appreciate the reviewer's insightful and constructive feedback. We have thoroughly addressed each of the comments and will revise the manuscript accordingly. Below, we provide detailed responses to all the points raised:*

Specific comments:

Line 20: Do you mean soil "sub"surface processes?

*Thank you for your careful reading. We appreciate the opportunity to clarify our meaning. In this context, we are referring to processes occurring on the surface of the soil particles, which are located throughout the soil profile. To clarify, the sentence in Line 20 has been revised to ensure this distinction is clear. The revised sentence reads:* **"Our findings conclusively demonstrate that the noninvasive electrical method allows real-time monitoring of calcite dissolution, serving as a direct cursor to the oxidation process of hydroquinone, enabling the observation of chemical interactions in soil solution, and on soil particle surfaces."**

Methods: I suggest removing the sub-section headings "2.1 materials" and "2.3 methods"

*We agree with the suggestion and will remove the sub-section headings as recommended.*

Line 95: Describe the geometric factor.

*We appreciate the comment. We use the relationship σ = G * Y to convert admittance (Y) to complex conductivity (σ), where G is the geometric factor. The geometric factor G (1/m) accounts for the geometry of the measurement setup, including electrode configuration and sample dimensions, enabling the conversion of measured electrical properties (admittance) to intrinsic material properties (complex conductivity). For our setup, G was determined by measuring the admittance of a series of electrolytes with*

*known electrical conductivities, as described in lines 104-107. The value of G for our measurements is 0.0127 $m^{-1}$.*

Line 92 – 98: The authors should also introduce alternative nomenclature e.g. real and imaginary conductivity, because this is a soil science audience. This way there is no ambiguity when readers compare with other literature.

*Thank you for the valuable suggestion. We agree that introducing alternative nomenclature will enhance clarity for a soil science audience, and we have revised the manuscript accordingly.*

Line 97: There are earlier references that determined the sensitivities of the quadrature and in-phase conductivities

*Thank you for pointing this out. We have updated the reference in Line 97 to Vinegar and Waxman (1984)*

Line 104: Was there a mesh preventing the sand sample from falling into the electrode casing?

*All electrodes are tightly inserted into the casing through a rubber band that matches the electrode's diameter, preventing sand from entering. Therefore, there is no mesh in the columns. The current electrodes extend through the entire sample, while the potential electrodes do not directly contact the sample but are connected through a clay mixture salt bridge. This configuration ensures no electrode polarization while maintaining secure contact with the sample.*

Lines 153 – 164: In Figure 2 you justify the "conservative" transport behavior of both hydroquinonens based on the breakthrough (C/C0 = 0.5) at 1 PV in panel (a). However, it is interesting that **the slope of the breakthrough curves differs between hydro- and benzo quinones.** The authors could compare their results to the Ogata-Banks solution for 1D conservative transport to highlight any deviations from the expected idealized behavior. **There appears to be a background concentration of benzoquinone in the sand-only column. Here the concentrations measurements also exhibit a large error spread when compared to the Mn-sand column**. The authors should address this in the text.

*Thank you for the insightful comment. We have revisited our data and evaluated the fit of our results to the advection-dispersion equation (ADE) solution, as described in "Soil Water Dynamics", chapter 7, page 307-310 (AW Warrick). The figure attached shows the fitting of the ADE solution to the breakthrough curves, indicating a good match for both hydroquinone and benzoquinone. The deviation in the benzoquinone results is likely due to measurement errors, as you noted. One possible factor contributing to these deviations could be the reduction of benzoquinone, given the natural conditions of the system in which the experiments were conducted. Such a setting inherently introduces some variability between runs, which may explain the larger standard deviations observed. We have highlighted these points in the revised text to provide greater clarity.*

[Figure]

Figure 2b shows that more benzoquinone exits the column as the hydroquinone. In line 120, I interpret that he Mn-sand col was only injected with hydroquinone, but the sand only with both. Is this correct? Please highlight this point more clearly in the text.

*Yes, that is correct. We added benzoquinone only to the sand-only columns to confirm that no significant reactions occurred with benzoquinone. The primary objective of the experiment was to follow the oxidation of hydroquinone. To clarify this in the manuscript, we have revised the text: "Upon reaching equilibrium, the inlet solution was replaced either by a mixed solution of hydroquinone and benzoquinone (0.1M each in CaCl₂ 5mM solution) for the sand-only columns or by a hydroquinone-only solution (0.1M in CaCl₂ 5mM solution) for the Mn-sandy soil columns. **The mixed solution was used for the sand-only columns to ensure no interactions occurred with benzoquinone, while the primary purpose was to follow the hydroquinone oxidation.** Both solutions passed through their respective columns for 4 or 8 pore volumes (PV), respectively, until mass balance was achieved."*

Line 172: Correct referencing error.

*Thank you for noticing. The reference issue at Line 172 has been fixed.*

Line 178 – 179: Why do you state that the Ca2+ concentrations were negligible? Your inflowing solution contained 5 mM Ca2+ and that is what you see in the outflow. Rather than being negligible the concentrations remained constant.

*Thank you for the comment. You are correct, the increase in Ca²⁺ is negligible, and we have revised the text accordingly. The updated sentence now reads: "This was accompanied by negligible concentrations of benzoquinone and manganese, while the Ca²⁺ concentration remained constant at 5 mM in the effluent."*

Line 182: How does the oxidation reaction change the surface charging properties of the MnO2? This is not clearly addressed in the paper.

*Thank you for your comment. In this line, we intended to indicate that the oxidation reaction leads to a reduction in available $MnO_2$ for further oxidation, as its concentration decreases. We address the surface properties of $MnO_2$ more explicitly later in the manuscript, particularly in Lines 238–239, where we discuss $MnO_2$ surface possible passivation due to $Mn^{2+}$ precipitation.*

Lines 190 – 204: Here the discussion presents a plausible conceptual model that describes the reactions taking place in the Mn-sand columns and their potential effects on SIP signatures. While the differences between control and treatment columns are apparent, the text discussion requires additional information: It would be an improvement for the manuscript to present the chemical reactions that the authors propose are occurring in the system, this would allow H+ changes to become immediately apparent.

*We added the following chemical reaction equation to Figure 2.*

Mass balance calculations: Currently the discussion remains very qualitative. The authors should check whether their mass balance supports their conclusions. For example, the extent aqueous Mn production, the total amount of outflow Ca2+ and pH change should be related to the stoichiometry of the chemical reactions. How much of the 4% CaCO3 is expected to be consumed based on the total breakthrough of calcite. Such an analysis would improve the plausibility of the conceptual model and highlight the potential of such monitoring schemes to deliver quantitative information. This would also further support the findings depicted in Figure 6.

*Thank you for the insightful comment. I will first describe the chemical reactions occurring in the column:*

*Calcite dissolution:*

(1) $CaCO_3 + H_2O \rightleftharpoons Ca^{2+} + HCO_3^- + OH^-$

*Manganese oxide reduction:*

(2) $MnO_2 + 4H^+ \rightleftharpoons Mn^{2+} + 2H_2O$

*Hydroquinone oxidation:*

(3) $C_6H_4(OH)_2 \rightleftharpoons C_6H_4O_2 + 2H^+ + 2e^-$

*Based on these reactions, I address your comments as follows:*

- **Mass balance**: *Although a complete mass balance is desirable, calculating the exact extent of produced $Mn^{2+}$ is challenging due to its likely precipitation within the system. While we have multiple measurements of pH in the column outlets, we hypothesize that the pH changes driving the cascade of reactions are mainly local, making it difficult to accurately correlate them with stoichiometric changes.*

- **Estimation of CaCO₃ dissolution**: *We estimated that CaCO₃ dissolution occurs in a 1:1 ratio with the release of $Ca^{2+}$ into the solution. Based on our measurements, approximately 0.26 g of CaCO₃ was washed out of the column. However, it is important to emphasize that this amount likely underestimates the CaCO₃ dissolved in the system. Some of the dissolved $Ca^{2+}$ may have been retained within the column or precipitated on the surfaces, making it difficult to determine the precise amount of CaCO₃ dissolved. Nevertheless, we have extensive supporting evidence from various measurements, which strongly supports our interpretation of CaCO₃ dissolution in this context.*